# Preliminary Study of the Effects of Eccentric-Overload Resistance Exercise on Physical Function and Torque Capacity in Chronic Kidney Disease

**DOI:** 10.3390/jfmk5040097

**Published:** 2020-12-18

**Authors:** Jared M. Gollie, Samir S. Patel, Joel D. Scholten, Michael O. Harris-Love

**Affiliations:** 1Physical Medicine and Rehabilitation and Research Services, Veterans Affairs Medical Center, Washington, DC 20422, USA; Jared.Gollie@va.gov; 2Department of Rehabilitation Science, George Mason University, Fairfax, VA 22030, USA; 3Department of Health, Human Function, and Rehabilitation Sciences, The George Washington University, Washington, DC 20006, USA; 4Renal Service, Veterans Affairs Medical Center, Washington, DC 20422, USA; Samir.Patel@va.gov; 5Department of Medicine, The George Washington University, Washington, DC 20006, USA; 6Physical Medicine and Rehabilitation Service, Veterans Affairs Medical Center, Washington, DC 20422, USA; Joel.Scholten@va.gov; 7Department of Physical Medicine and Rehabilitation, University of Colorado, Aurora, CO 80045, USA; 8Geriatric Research Education and Clinical Center, VA Eastern Colorado Healthcare System, Aurora, CO 80045, USA

**Keywords:** muscle fatigue, renal failure, eccentric, flywheel, sit-to-stand, strength

## Abstract

The purpose of this preliminary study was to describe changes in physical function and torque capacity in adults with chronic kidney disease (CKD) in response to a novel progressive eccentric-overload resistance exercise (ERE) regime. Participants included men (*n* = 4) diagnosed with CKD according to estimated glomerular filtration rate (eGFR) between 59 and 15 mL/kg/1.73 m^2^ and not requiring dialysis. Physical function was determined by the Short Physical Performance Battery (SPPB), five repetitions of a sit-to-stand (STS) task, and timed-up and go (TUG). Knee extensor strength was assessed using both isometric and isokinetic contractions and performance fatigability indexes were calculated during a 30-s maximal isometric test and a 30-contraction isokinetic test at 180°/second. None of the patients exhibited significant worsening in their health status after training. Participants demonstrated improvements in several measures of physical function and torque capacity following 24 sessions of ERE. Following training, performance fatigability remained relatively stable despite the increases in torque capacity, indicating the potential for greater fatigue resistance. These findings provide initial evidence for ERE as a potential treatment option to combat declines in physical function and neuromuscular impairments in people with CKD. Future research is required to determine optimal progression strategies for maximizing specific neuromuscular and functional outcomes when using ERE in this patient population.

## 1. Introduction

Chronic kidney disease (CKD) is defined as decreased kidney function and is clinically diagnosed with an estimated glomerular filtration rate (eGFR) of less than 60 mL/min per 1.73 m^2^ or markers of kidney damage, or both, of at least 3 months duration [1]. Mild-to-severe kidney dysfunction is classified as those exhibiting eGFR between 59 and 15 mL/min per 1.73 m^2^ (i.e., CKD stages 3 and 4) while an eGFR of less than 15 mL/min per 1.73 m^2^ identifies those with kidney failure, requiring renal replacement therapy [1]. CKD is most prevalent in adults 65 years of age and older, with the two leading causes being diabetes mellitus and hypertension [1]. The health consequences of CKD are underscored by evidence of greater risk for mortality, disability, and activity limitations when compared to those without CKD [2,3,4].

The pathophysiology of CKD is complex and systemic in nature, affecting multiple organ systems [5]. The neuromuscular system is especially susceptible to secondary consequences of renal insufficiency due to the accumulation of uremic toxins [6,7,8,9,10,11,12]. This includes the promotion of skeletal muscle atrophy resulting from an upregulation of protein degradation and downregulation of protein synthesis [13]. Additionally, uremic metabolites have been identified to impair skeletal muscle mitochondrial oxidative energy production [14]. Moreover, reduced axonal and skeletal muscle membrane excitability and skeletal muscle sodium-potassium pump inhibition have also been shown to be impaired in those with CKD [11,12,15,16,17]. The combination of reduced torque-generating capacity with compromised bioenergetic processes places individuals with CKD at a greater risk for activity limitations and excessive levels of fatigability. Thus, treatment approaches capable of enhancing torque-generating capacity and functional capabilities while simultaneously reducing fatigability would be advantageous for this patient population.

Eccentric-overload resistance exercise (ERE) uses inertial loading to allow for optimal muscle loading through the entire concentric muscle action [18,19]. While not conclusive [20], evidence suggests that ERE may offer a superior stimulus for increasing concentric and eccentric strength, muscle power, muscle hypertrophy, vertical jump height, and running speed as compared to traditional resistance exercise [21,22]. Moreover, ERE is shown to counteract the negative metabolic consequences associated with neuromuscular unloading by maintaining skeletal muscle oxidative properties [23]. While these findings posit ERE as a potential treatment option for addressing co-existing neuromuscular impairments and activity limitations simultaneously, to our knowledge, no studies have examined the application of ERE in adults with CKD. Therefore, the purpose of this preliminary study was to describe changes in physical function and torque capacity in adults with CKD not on dialysis in response to a novel progressive ERE regime.

## 2. Materials and Methods

### 2.1. Participant Characteristics and Ethical Approval

Community-dwelling male veterans were screened and referred by the Renal Clinic staff at the Veterans Affairs Medical Center in Washington D.C. (DC VAMC) for potential enrollment. Patients who were ambulatory, with or without use of an assistive device, aged 18−85 years, diagnosed with CKD stages 3−5 (i.e., eGFR <59 mL/min per 1.73 m^2^ body surface area (BSA)), and not on dialysis were admitted [24]. Exclusion criteria included a history of acute kidney injury, any uncontrolled cardiovascular or musculoskeletal problems, or having a pacemaker or implantable cardioverter defibrillator. This study was approved by the DC VAMC Institutional Review Board (IRB) and Research & Development Committee’s (Protocol #01903) on 27 February 2018. All participants voluntarily provided written informed consent using an IRB-approved form prior to study participation.

Data were collected by trained staff in the Renal Clinic and Human Performance Research Unit at the DC VAMC. Each patient reported to the Human Performance Research Unit in the Clinical Research Center at the DC VAMC to complete neuromuscular strength and physical function assessments lasting for a total of 60−90 min. Patients were asked to refrain from any vigorous activity for at least 24 h prior to testing. Function testing was conducted using the Short Physical Performance Battery (SPPB) and Timed-Up and Go (TUG) tests, and preceded strength testing. Peak knee extensor strength of the dominant leg was determined using maximal voluntary isometric contractions (MVICs), and isokinetic knee extensor contractions at 180°/s and 60°/s. Performance fatigability was assessed during both isometric contractions lasting 30 s in duration and in response to 30 repeated knee extension contractions at 180°/s, as previously described [17].

### 2.2. Eccentric-Overload Resistance Exercise (ERE)

Exercise intensity (i.e., inertial load) when using a flywheel device to promote eccentric-overload is determined by the mass, radius, and angular acceleration of the flywheel [18,25]. Exercise studies investigating ERE most commonly use a protocol consisting of four sets of seven repetitions performed at maximal effort [18]. However, when working with older adults and clinical populations, performing muscular contractions at maximal effort may not be safe or well tolerated. To ensure participant safety while maximizing neuromuscular and functional adaptations, the ERE program designed for this study incorporated principles of individualization, specificity, progression overload, and variation [26]. Inertial load and contraction velocity were systematically manipulated throughout the duration of training. The first four training sessions were used as a familiarization period to acclimate the participants with the flywheel device and to identify the individualized inertial load and movement velocity to be used (Exxentric kBox4 Pro, Bromma, Sweden) [23]. The starting inertial load for each participant was determined by identifying the load that maximized average concentric power output at a self-selected movement velocity and could be performed for 12 repetitions. The next twenty training sessions were organized into three mesocycles (Table 1). Mesocycle 1 (sessions 5 to 8) demarcates the start the actual exercise regime with the established inertial load and velocity. The start of Mesocycle 2 (session 9) and Mesocycle 3 (session 17) signifies an increase in inertial load and the re-establishment of the new self-selected movement velocity. Standardized increases in movement velocity were prescribed every other session. The systematic manipulation of both inertial load and movement velocity ensured continued increases in concentric and eccentric power output throughout the duration of the exercise program (Figure 1). Prior to each exercise session, patients completed a warm-up on either a recumbent bike or treadmill at a self-selected intensity maintained for 10 min. The ERE regime included squat, shoulder press, seated row, and bicep curl exercises, though only data on the squat are reported herein.

### 2.3. Physical Performance Measures

The SPPB is a measure used to assess lower extremity function composed of tasks assessing balance, walking speed, and the ability to rise from a chair and has been shown to be predictive of subsequent disability in persons over 70 years of age [27]. A composite score is generated with the highest possible score being 12. Those completing the respective tasks were assigned a score of 1 to 4, with higher scores representative of better performance of the task. Participants unable to complete a task were given a score of zero for that task. Complete details of the SPPB have been described elsewhere [28]. The sit-to-stand (STS) task was determined as the time taken to complete five sit-to-stand repetitions. Gait speed was calculated using the time taken to walk 4 m as a usual walking pace. The TUG test consisted of the time required by each patient to rise from a chair, walk 3 m, turn around, walk back to the chair, and finish in the seated position.

### 2.4. Maximal Voluntary Isometric Contraction (MVIC) and Isokinetic Testing

All strength testing was performed using Biodex System 4 Pro Dynamometer (Biodex Medical, Shirley, NY, USA). Participants were placed in the seated position with the axis of rotation of the dynamometer’s lever arm aligned with the axis of rotation of the participant’s knee. Extraneous movements were restricted by securing the participant to the seat of the dynamometer using a four-point harness. Participants were allowed to use their hands to grasp the bars of the dynamometer to aid with body stabilization during testing.

A familiarization session was provided before data collection to orient each participant to the testing procedures. Participants were instructed to extend or kick their leg as “hard” as possible following the test administrator’s requests. During MVIC testing, participants performed a sustained isometric contraction for 5 s while being provided strong verbal encouragement. Three repetitions were completed, ensuring that each repetition was within 10% of each other. A rest period of approximately 60 s was provided between MVIC efforts. Similarly, peak isokinetic knee extension torque at 180°/s and 60°/s was determined by completing five continuous isokinetic knee extensor contractions at the respective resistances. Peak isokinetic torque was determined as the average of the three highest torque values achieved. Pre-contraction tension and countermovement were removed by test administrators and the dynamometer was zeroed just before initiation of contraction.

### 2.5. Performance Fatigability

Performance fatigability was assessed as changes in torque of the dominant leg during isometric and isokinetic muscle contractions. Similar procedures used during the MVIC and isokinetic strength testing were used for the performance fatigability assessments. For isometric performance fatigability, patients were asked to perform an MVIC lasting 30 s in duration. Isokinetic performance fatigability was assessed using a protocol consisting of 30 repetitions or until failure, whichever came first, at a resistance of 180°/s. Strong verbal encouragement was provided by the test administrator and visual feedback of the torque curve was displayed on a computer monitor throughout the duration of the test to provide knowledge of performance information to the patient. Performance fatigability indexes (FIs) for isometric and isokinetic testing were quantified as the percent reduction in torque over time (Equation (1)). For isometric FI, initial torque was determined as the highest torque achieved within the first 5 s of the MVIC and final torque was determined as the torque at the termination of the test. For isokinetic FI, initial torque was calculated by averaging the three highest torque values achieved within the first 5 completed contractions, while final torque was calculated as the average of the last 3 completed contractions of the test.
Fatigability Index (FI) = final torque/initial torque × 100(1)

### 2.6. Statistical Analysis

Descriptive statistics were used due to the preliminary nature of the study. All values are expressed as mean (range) unless otherwise noted. Group differences between pre-test and post-test values in physical function, strength and performance fatigability following 24-session of ERE are presented as percent change. Individual change in physical function, strength and performance fatigability were visually depicted using scatter plots.

## 3. Results

### 3.1. Participant Demographics

Participant characteristics are summarized in Table 2. Participants were representative of the vast majority of older patients with CKD not on dialysis. Two participants were classified as CKD stage 3a, one participant as CKD stage 4, and one participant as CKD stage 5. All participants had diabetic and/or hypertensive kidney disease, with one participant also experiencing macro- and microvascular disease and claudication of the extremities. Participants were ambulatory with controlled hypertension and blood glucose. ERE was well tolerated with no adverse events reported. No change in health status or clinical representation was observed as a result of ERE. Additionally, participants did not report experiencing excessive muscle soreness or discomfort commonly associated with unaccustomed exercise. The four participants were able to complete all 24 sessions of ERE; however, the timeframe varied between 12 and 30 weeks. The factors contributing to the differences in time to complete 24 sessions of ERE seemed to be primarily related to time missed due to federal holidays. Several sessions were also cancelled due to elevations in resting blood pressure deemed to be unsafe for engaging in exercise in the participant with CKD stage 5.

### 3.2. Physical Function and Torque Capacity

Mean change in physical function and torque capacity are presented in Table 3. Three of the four participants showed improvements in SPPB of one point and STS performance (Figure 2). TUG performance also improved in two participants, with no change in the other two participants (individual data not shown). Gait speed was slightly slower after ERE in three participants, with no change in gait speed observed in one participant (individual data not shown). Three of the four participants experienced increases in isometric and isokinetic torque capacity following ERE (Figure 3). Fatigability status remained relatively stable in three participants, with one participant experiencing improvements in both isometric and isokinetic fatigability assessments after ERE. When comparing performance fatigability assessments, participants were generally more fatigable during the isokinetic as compared to the isometric assessment before and after training (Figure 4).

## 4. Discussion

This preliminary study demonstrated that the ERE regimen described here was safe and feasible in adults with CKD stages 3−5 not on dialysis. None of the participants exhibited significant worsening in their health status as determined by clinical evaluation or by assessments of kidney function as per clinical practice guideline recommendations [24]. Participants demonstrated improvements in several measures of functional status and torque capacity following 24 sessions of ERE. Importantly, performance fatigability remained relatively stable despite increases in torque capacity, indicating greater fatigue resistance after ERE. Taken together, these findings provide initial evidence for ERE as a potential treatment option in people with CKD to combat declines in physical function and neuromuscular impairments.

Measures of physical function have been shown to be predictive of mortality, disability, and adverse health outcomes [27,29,30,31,32,33,34]. Progressive declines in renal function are associated with worsening in physical function [3,35,36]. The functional status of the participants included in this study was diminished at baseline. Three of the four participants had SPPB scores of ≤10, suggesting the potential for risk of mobility disability [37]. The mean gait speed of 1.0 m/s was ≈26.8% slower than previously reported age-predicted normative values and categorizes these individuals as having intermediate or “mildly abnormal” gait speed [38,39]. Similarly, sit-to-stand time at baseline was 34.2% slower than age-predicted norms [40].

Investigations examining the application of resistance exercise using flywheel devices in older adults report increases in physical functioning, torque, power, neural activation, and tendon stiffness [41,42,43]. After 24 sessions of ERE, three of the four participants improved their SPPB performance by one point, meeting the minimally clinically important difference for this outcome measure [44]. In addition, times to complete the STS and TUG assessments were reduced following training by an average of 15.8% and 6.1%, respectively. Furthermore, improvements in knee extensor isometric and isokinetic strength of 8%, 14.8% (180°/s), and 7.8% (60°/s) were observed after training. Despite these improvements, however, customary gait speed was slower after training. Therefore, while these preliminary results provide promising findings for the application of ERE in CKD, future large-scale studies are needed to determine the efficacy of ERE as well as the appropriateness of ERE for addressing gait deficits.

Performance fatigability, when measured using maximal contractions, has been shown to increase after resistance exercise such that the increases in maximal force capacity result in greater levels of fatigability [45,46]. This phenomenon is described by the force–fatigability relationship where the extent to which performance fatigability occurs is directly related to the amount of neuromuscular force exerted during a given task [47]. Thus, increases in force capacity following resistance exercise interventions may result in greater performance fatigability when assessed using maximal contractions. Conversely, the lack of change in performance fatigability despite increases in torque capacity may be viewed as a positive adaptation following resistance exercise [45]. In the present study, mean changes in performance fatigability showed slight improvements in both isometric and dynamic measures. However, this improvement was primarily driven by the changes observed in one participant (Figure 4). The performance fatigability response profiles in the other three participants are consistent with the expected changes in torque during isometric and dynamic fatigability protocols [17,48]. Moreover, isokinetic performance fatigability was greater compared to isometric fatigability in these individuals, as previously reported [49].

Interestingly, the performance fatigability response profiles in one participant are uncharacteristic of what is typically observed. The increases in torque over 30 isokinetic contractions performed at 180°/sec suggest that this participant did not fatigue during this protocol. Similarly, minimal change in maximal isometric torque was observed in this individual during the isometric fatigability test. The potential neural and morphological mechanisms contributing to the lack of fatigue experienced during these protocols is unclear. One potential explanation for the observations of this participant during the isokinetic fatigability protocol is the low torque-generating capacity (pre = 39 ft-lbs; post = 31.5 ft-lbs). Seminal studies conducted by Thorstensson and colleagues, examining performance fatigability during repeated dynamic contractions, found that the rate of fatigue was related to skeletal muscle characteristics [50]. The reliance on low-threshold motor units and, thus, slow twitch muscle fibers or the inability to recruit high-threshold motor units (and fast twitch muscle fibers) at the contraction speed of 180°/sec, therefore, may have contributed to the fatigue resistance noted. Importantly, this participant was the only participant to have vascular disease (macro and micro) as well as claudication of the lower extremity. Thus, this individual may have intentionally limited his effort during testing due to sensations associated with volitional activity which may have also contributed to the lower torque capacity.

To date, optimal prescription of ERE for enhancing specific neuromuscular and functional outcomes has not been fully established for clinical populations. However, understanding the interactions between inertial load and movement velocity seem critical for enhancing strength and power using flywheel devices. For example, Martinez-Aranda et al. [51] observed that while force increased with progression of inertial loads up to 0.0375 kg⋅m^−2^ in younger men, power output was found to decline. Similarly, movement velocity in the squat exercise using ERE showed a linear reduction across inertial loads of 0.010 to 0.100 kg⋅m^−2^ [52]. The ability to capitalize on the unique stimulus provided by flywheel devices requires the individual to perform the concentric portion of the movement at higher velocities to increase the potential for greater eccentric-overload [18,25]. Regarding maximizing power outputs for concentric and eccentric phases of the squat exercise, a load of 0.025 kg⋅m^−2^ was found to elicit greater concentric power output, while the greatest eccentric overload was achieved at a load of 0.075 kg⋅m^−2^ in high-level handball players [53]. The findings of the present study demonstrate that inertial load and movement velocity can safely be progressed in patients with CKD. The inertial loads ranged from 0.02 to 0.06 kg·m^−2^, promoting progressive increases in force generation across each mesocycle. Furthermore, by increasing movement velocity within each mesocycle, greater concentric and eccentric power outputs were achieved throughout the duration of training.

Resistance exercise has been shown to be safe and efficacious in adults with CKD [6,54,55,56,57,58]. However, much of this evidence stems from studies conducted on individuals with end-stage kidney disease, with limited information available in people with CKD not requiring dialysis [59,60]. Adults with mildly-to-severely decreased kidney function (i.e., eGFR 59−15 mL/min per 1.73 m^2^) are at greater risk for mortality caused by cardiovascular disease as well as transitioning to end-stage kidney disease [61,62]. Therefore, structured and individualized exercise interventions are recommended for patients with mild-to-severe CKD for enhancing physical function [63]. ERE may offer one treatment option for addressing neuromuscular impairments and activity limitations in those with CKD not on dialysis. The regimen described in the present study outlines how both inertial load and movement velocity can be safely progressed based on with the goal of enhancing power output and physical function.

There are several limitations to this study, including the small number of observations. This greatly limits the ability to generalize the current findings, and therefore, the results of this study should be interpreted with caution. However, these data provide preliminary evidence for the potential application of ERE in CKD patients. In addition to the number of observations, no established protocol for implementing ERE in clinical populations currently exists. This makes it difficult to determine how to best apply ERE for enhancing function and neuromuscular characteristics. Moreover, it is unclear if the magnitude of the response to the eccentric-overload stimulus is similar in clinical populations as healthy younger populations or if ERE provides greater benefit than traditional resistance exercise approaches. Therefore, future research should focus on determining optimal progression strategies for ERE for maximizing specific outcomes and comparing ERE to traditional resistance exercise in patients with CKD, as well as other clinical populations.

## 5. Conclusions

Resistance exercise approaches emphasizing eccentric-overload may provide an additional treatment option for improving neuromuscular force outcomes and physical function in adults with CKD not requiring dialysis. The findings presented here provide preliminary evidence for the safety and feasibility of the ERE program detailed in this study. Future research is required to determine optimal progression strategies for ERE for maximizing specific neuromuscular, functional, and health outcomes.

## Figures and Tables

**Figure 1 jfmk-05-00097-f001:**
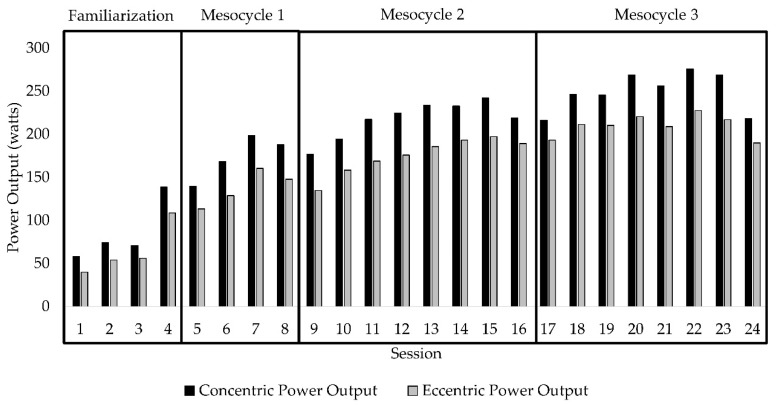
Change in mean concentric and eccentric power output in the squat exercise.

**Figure 2 jfmk-05-00097-f002:**
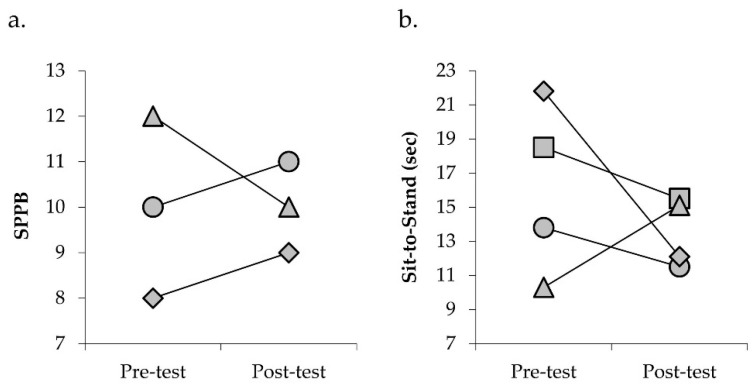
Scatter plots depicting individual changes in SPPB (**a**) and five repetitions of the sit-to-stand task (**b**) following 24 sessions of eccentric-overload resistance exercise in patients with chronic kidney disease stages 3−5 not on dialysis. Participant 2 and participant 3 each had an SPPB pre-test score of 8 and a post-test score of 9; therefore, only 3 data points are depicted in the graph (**a**). Circle, participant 1; square, participant 2; diamond, participant 3; triangle, participant 4. SPPB, Short Physical Performance Battery; sec, seconds.

**Figure 3 jfmk-05-00097-f003:**
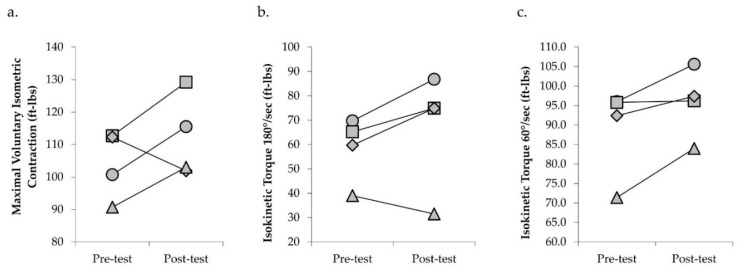
Scatter plots depicting individual changes in knee extensor maximal voluntary isometric contraction torque (**a**), isokinetic torque at 180°/sec (**b**), and isokinetic torque at 60°/sec (**c**) following 24 sessions of eccentric-overload resistance exercise in participants with chronic kidney disease stages 3−5 not on dialysis. Circle, participant 1; square, participant 2; diamond, participant 3; triangle, participant 4.

**Figure 4 jfmk-05-00097-f004:**
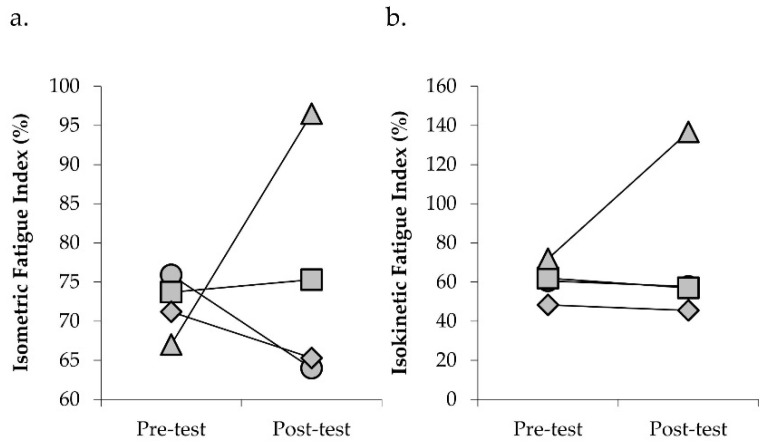
Scatter plots depicting individual changes in fatigability index during isometric (**a**) and isokinetic (**b**) knee extensor contractions following 24 sessions of eccentric-overload resistance exercise in participants with chronic kidney disease stages 3−5 not on dialysis. Circle, participant 1; square, participant 2; diamond, participant 3; triangle, participant 4.

**Table 1 jfmk-05-00097-t001:** Description of eccentric-overload resistance exercise for adults with chronic kidney disease (CKD).

	Familiarization	Mesocycle 1	Mesocycle 2	Mesocycle 3
Sessions	1−4	5−8	9−16	17−24
Volume (sets × reps)	2 × 12	3 × 12	3 × 12	3 × 12
Mean Intensity (inertial load kg⋅m^−2^)	N/A	0.028	0.036	0.045
Rest between sets	60 s	60 s	60 s	60 s

Abbreviations: kg⋅m^−2^, kilograms per meter squared; s, seconds; N/A, not applicable.

**Table 2 jfmk-05-00097-t002:** Participant characteristics (*n* = 4).

	Pre-Test	Post-Test	Reference Range
**Demographics**			
Age (years)	68.8 (57−78)		
Sex (*n*)	Male (*n* = 4)		
Body Mass Index (BMI) (kg/m^2^)	34.1 (24.5−40.8)	34.3 (23.6−40.3)	
Blood Pressure Medication (*n*)	2.8 (2.0−4.0)	2.8 (2.0−4.0)	
**Laboratory Values**			
eGFR (mL/kg/1.73 m^2^ BSA)	37.6 (13.3−55.2)	38.0 (14.3−59.8)	>60.0
Hemoglobin (Hgb) (g/dL)	12.1 (10.7−13.6)	11.9 (10.3−13.6)	13.2−17.3
Potassium (K) (mEq/L)	4.4 (4.0−4.7)	4.6 (4.1−5.1)	3.5−5.3
Bicarbonate (HCO3) (mEq/L)	26.0 (24.0−29.0)	26.3 (21.0−31.0)	21.0−31.0
Blood Urea Nitrogen (BUN) (mg/dL)	33.3 (10.0−69.0)	35.8 (10.0−61.0)	6.0−23.0
Creatinine (mg/dL)	2.8 (1.5−5.1)	2.8 (1.4−4.8)	0.7−1.5
Calcium (Ca) (mg/dL)	9.0 (7.8−9.8)	9.1 (7.9−10.1)	8.9−10.5
Albumin (Alb) (g/dL)	4.2 (3.6−4.7)	4.1 (3.6−4.6)	3.7−5.0

Abbreviations: BMI, body mass index; eGFR, estimated glomerular filtration rate; Hgb, hemoglobin; K, Potassium; HCO3, bicarbonate; BUN, blood urea nitrogen; Ca, calcium; Alb, albumin; kg/m^2^, kilograms per meter; mL/kg/1.73m^2^ BSA, milliliter per kilogram per 1.73 m body surface area; g/dL, grams per deciliter; mEq/L, milliequivalent per liter; mg/L, milligram per deciliter. Data are presented as mean (range).

**Table 3 jfmk-05-00097-t003:** Change in physical function and torque capacity.

	Pre-Test	Post-Test	% Change
**Functional Outcomes**			
Short Physical Performance Battery (SPPB)	9.5 (8−12)	9.8 (9−11)	+2.6
Five repetitions of Sit-to-Stand (STS) (s)	16.1 (10.3−21.8)	13.6 (11.5−15.5)	−15.8
Gait speed (m/s)	1.0 (0.8−1.5)	0.90 (0.7−1.2)	−10.1
Timed Up and Go (TUG) (s)	11.5 (9.6−13.8)	10.8 (8.6−13.3)	−6.1
**Strength Outcomes**			
Knee Extensor Isometric Strength (ft-lbs)	104.1 (90.7−112.7)	112.4 (101.9−129.2)	+8.0
Knee Extensor Isokinetic Strength (180°/s) (ft-lbs)	58.4 (39.0−69.7)	67.0 (31.5−86.8)	+14.8
Knee Extensor Isokinetic Strength (60°/s) (ft-lbs)	88.9 (71.4−96.0)	95.8 (84.0−105.6)	+7.8
**Performance Fatigability Outcomes**			
Isometric FI (%)	72.2 (67.0−75.9)	75.9 (64−96.5)	+4.6
Isokinetic FI (%)	69.2 (48.3−71.9)	74.8 (45.5−136.7)	+22.3

Abbreviations: SPPB, Short Physical Performance Battery; STS, Sit-to-Stand; FI, fatigability index; s, seconds; m/s, meters per second; ft-lbs, foot-pounds; %, percent. Data are presented as mean (range).

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
