# Peer review of "Preliminary Study of the Effects of Eccentric-Overload Resistance Exercise on Physical Function and Torque Capacity in Chronic Kidney Disease"

_jfmk, 2020, doi:10.3390/jfmk5040097_

Round 1

Reviewer 1 Report

This is a case series aimed to examine a progressive eccentric overload resistance exercise (ERE) regime training on physical function and torque capacity in patients with chronic kidney disease (CKD). It shows improved torque and functional capacity, including greater fatigue resistance in three out of four examined CKD patients. Based on this, the authors conclude that ERE could be considered as a potential treatment option to ameliorate changes in physical function and neuromuscular impairments in patients with CKD.

Although being well-conducted and described, the study’s major limitation is obviously related to the number of patients and their overall functional and morbid status, as also acknowledged by the authors. Secondly, although it examined several functional parameters, the study could have included additional measurements like the postural stability (related to risk for falls) and self-reported everyday-life physical functions potentially adding clinical relevance and importance to the study. In that setting measuring also cognitive functioning would be necessary. Considering these limitations, the study should be pitched as a proof of concept and elaborate briefly on potential clinical implications and future directions.

Following the JFMK editorial on the utility of eccentric training (unfortunately not cited by the authors), it is a reasonable step in promoting the use of isoinertial technology in the realm of rehabilitation.

I have only one minor suggestion related to the presentation of data on physical function and torque capacity (Table 3). Namely, considering the small number of patients, data should be given as mean and range rather than SD.

Author Response

This is a case series aimed to examine a progressive eccentric overload resistance exercise (ERE) regime training on physical function and torque capacity in patients with chronic kidney disease (CKD). It shows improved torque and functional capacity, including greater fatigue resistance in three out of four examined CKD patients. Based on this, the authors conclude that ERE could be considered as a potential treatment option to ameliorate changes in physical function and neuromuscular impairments in patients with CKD.

Although being well-conducted and described, the study’s major limitation is obviously related to the number of patients and their overall functional and morbid status, as also acknowledged by the authors.

  • We thank the reviewer for this comment. While the small number of participants is a major limitation to this work as indicated we believe this information is valuable for understanding the potential applications of eccentric-focused resistance exercise in clinical settings and provides the necessary information supporting future large-scale studies.

Secondly, although it examined several functional parameters, the study could have included additional measurements like the postural stability (related to risk for falls) and self-reported everyday-life physical functions potentially adding clinical relevance and importance to the study. In that setting measuring also cognitive functioning would be necessary.

  • We thank the reviewer for this comment and agree that the inclusion of additional measures would have strengthened the current study. Unfortunately, the addition of the described measures is not possible at this time. However, we plan to use this information to inform future larger studies of which strong consideration will be given for the inclusion of the proposed measures.  

Considering these limitations, the study should be pitched as a proof of concept and elaborate briefly on potential clinical implications and future directions.

  • We thank the reviewer for this comment and have changed the title to identify the study as a “preliminary study” as opposed to a “case series”. We have also added a paragraph on the importance of initiating exercise during earlier stages of CKD (please see lines 632-640 of the marked-up version). We briefly described future directions we feel are of importance following this work in our original submission and therefore have not added any addition information specific to this comment (please see lines 658-662) as well as throughout the Discussion (please see lines

Following the JFMK editorial on the utility of eccentric training (unfortunately not cited by the authors), it is a reasonable step in promoting the use of isoinertial technology in the realm of rehabilitation.

  • We thank the reviewer for bringing this Editorial to our attention and have added this reference to our paper.

I have only one minor suggestion related to the presentation of data on physical function and torque capacity (Table 3). Namely, considering the small number of patients, data should be given as mean and range rather than SD.

  • We thank the reviewer for this suggestion and have changed the values in Table 3 from mean (SD) to mean (range).

Reviewer 2 Report

This is a preliminary study of ERE exercise in patients with stage 3-5 CKD not requiring dialysis.  This study shows the safety of these ERE exercise in this group of patients--especially when begun at moderate intensity.  The paper is well written with excellent background and discussion sections.  I have only a few coments/sggestions:

1.  I think it is important in the title and the abstract/introduction to state that this is a preliminary study which is more informative to the reader than a  'case series'.

2.  The descriptions of the individual patients are interesting, but it is very hard (at least for me).  I think, however, that selected results would be best presented in a table. Similarly, there do not appear to be clinically many meaningful changes in any of the renal function or metabolic parameters pre to post test, so simplification would be helpful.

3.  My only problems with the design of the study are the marked variation between study subjects chosen and control for the etiology of the CKD.  Patients with CKD due to an inflammatory process (e.g. autoimmune disease) may be qujite different from one with CKD due to hypertension.  Also,  there is increased sarcopenia with progressive CKD and separately with age--s0 using each patient as his own control is reasonable--but comparing a 57 year old to a 75 year old may not be.  Similarly, it would be advisable in the future to suggest some measurement of pre-test activity  level in terms of expected improvements.  The anticipated changes in strength/fatigibility might be different in the stage 3 CKD patient who walks 2 miles a day from the CKD 5 patient who is sedentary.

Small point--why are lines 236-45 in italics?

Author Response

This is a preliminary study of ERE exercise in patients with stage 3-5 CKD not requiring dialysis.  This study shows the safety of these ERE exercise in this group of patients--especially when begun at moderate intensity.  The paper is well written with excellent background and discussion sections.  I have only a few comments/suggestions:

  1. I think it is important in the title and the abstract/introduction to state that this is a preliminary study which is more informative to the reader than a 'case series'.
  • We thank the reviewer for this comment and have changed the title to indicate this study as a preliminary study rather than a case series.
  1. The descriptions of the individual patients are interesting, but it is very hard (at least for me).  I think, however, that selected results would be best presented in a table. Similarly, there do not appear to be clinically many meaningful changes in any of the renal function or metabolic parameters pre to post test, so simplification would be helpful.
  • We thank the reviewer for this suggestion and have modified our Results Section (please see lines 184-213 of the marked-up version).
  1. My only problems with the design of the study are the marked variation between study subjects chosen and control for the etiology of the CKD.  Patients with CKD due to an inflammatory process (e.g. autoimmune disease) may be quite different from one with CKD due to hypertension.  Also, there is increased sarcopenia with progressive CKD and separately with age--so using each patient as his own control is reasonable--but comparing a 57 year old to a 75 year old may not be.  Similarly, it would be advisable in the future to suggest some measurement of pre-test activity level in terms of expected improvements.  The anticipated changes in strength/fatigability might be different in the stage 3 CKD patient who walks 2 miles a day from the CKD 5 patient who is sedentary.
  • We thank the reviewer for raising this concern. As the reviewer adequately points out, the etiology of CKD is complex with several secondary consequences further compounding the challenges experienced by this population especially during the later stages of kidney disease. We feel that the participants described in our study are representative of the majority of patients with stages 3-5 CKD not requiring dialysis and reflect the disease spectrum and heterogeneity across patients commonly experienced in clinical practice. We can confirm that none of the patients had CKD due to inflammatory process and all patients had hypertensive and or diabetic kidney disease with one patient also experiencing macro and micro vascular disease and claudication of the extremities (57 year-old) (please see lines 186-188 and 617-620). Importantly, despite the inclusion of patients with differing stages of CKD not on dialysis, the exercise intervention was safe, feasible, and well-tolerated with positive effects on strength, fatigability, and physical function. We view these findings to be extremely valuable for this clinical population given the limited treatment options.
  • We thank the reviewer for this suggestion and agree that the addition of an activity level measure would strengthen any further work. We will strongly consider including such a measure in future studies we may design.

Small point--why are lines 236-45 in italics?

  • We thank the reviewer for bringing this to our attention. We have corrected this issue.